# How high energy fluxes may affect Rayleigh–Taylor instability growth in young supernova remnants

C.C. Kuranz[1], H.-S. Park[2], C.M. Huntington[2], A.R. Miles[2], B.A. Remington[2], T. Plewa[3], M.R. Trantham[1], H.F. Robey[2], D. Shvarts[4,5], A. Shimony[4,5], K. Raman[2], S. MacLaren[2], W.C. Wan [1,6], F.W. Doss[6], J. Kline[6], K.A. Flippo[6], G. Malamud[1,5], T.A. Handy[1], S. Prisbrey[2], C.M. Krauland[7], S.R. Klein[1], E.C. Harding[8], R. Wallace[2], M. J. Grosskopf[9], D.C. Marion[1], D. Kalantar[2], E. Giraldez[7] & R.P. Drake [1]

Energy-transport effects can alter the structure that develops as a supernova evolves into a supernova remnant. The Rayleigh–Taylor instability is thought to produce structure at the interface between the stellar ejecta and the circumstellar matter, based on simple models and hydrodynamic simulations. Here we report experimental results from the National Ignition Facility to explore how large energy fluxes, which are present in supernovae, affect this structure. We observed a reduction in Rayleigh–Taylor growth. In analyzing the comparison with supernova SN1993J, a Type II supernova, we found that the energy fluxes produced by heat conduction appear to be larger than the radiative energy fluxes, and large enough to have dramatic consequences. No reported astrophysical simulations have included radiation and heat conduction self-consistently in modeling supernova remnants and these dynamics should be noted in the understanding of young supernova remnants.

[1] University of Michigan, Ann Arbor 48109 Michigan, USA. [2] Lawrence Livermore National Laboratory, Livermore 94550 California, USA. [3] Florida State University, Tallahassee 32306 Florida, USA. [4] Ben Gurion University of the Negev, Be'er-Sheva 84015, Israel. [5] Nuclear Research Center Negev, Be'er Sheva 84190, Israel. [6] Los Alamos National Laboratory, Los Alamos 87545 New Mexico, USA. [7] General Atomics, San Diego 92186 California, USA. [8] Sandia National Laboratory, Albuquerque 87185 New Mexico, USA. [9] Simon Fraser University, Burnaby, BC, Canada. Correspondence and requests for materials should be addressed to C.C.K. (email: ckuranz@umich.edu)

When a blast wave emerges from an exploding star, it drives a forward shock into the circumstellar medium (CSM) and a reverse shock forms in the expanding stellar ejecta, creating a young supernova remnant (SNR). As mass accumulates in the shocked layers, the interface between these two shocks decelerates, becoming unstable to the Rayleigh–Taylor (RT) instability. Simulations[1] predict that RT produces structures at this interface, having a range of spatial scales. When the CSM is dense enough, as in the case of SN1993J where the ejecta density has a steep density gradient compared with other Type II supernovae, the hot shocked matter can produce significant radiative fluxes that affect the emission from the SNR and potentially alter the behavior of the RT[2]. Standard models predict that the reverse shock heats the incoming ejecta to several hundred electronvolts, leading to the formation[2] of a cool, dense layer of shocked ejecta, perhaps enhanced by a layer of collapsed ejecta formed before shock breakout[3–5]. As this dense layer expands, it eventually becomes transparent enough that the energy radiated from the reverse shock penetrates and ionizes the dense material. The radiation heats the shocked layer during this penetration, via the photo-electric effect and eventually the RT-unstable surface.

RT growth may be reduced or quenched by high-energy fluxes that cause the removal of material (ablation) from an unstable interface. In the case of SNRs, radiation is incident from within the shocked ejecta and so will only affect the interface during the phase when the dense layer at the interface becomes partially transparent. In contrast, the conductive fluxes from the CSM through the interface are present continually. They are large enough that they might fundamentally change the commonly assumed structure of this region. This motivates future experimental and theoretical studies. We first analyzed this aspect of

SN1993J using the models that Suzuki et al.[6] and Fransson et al.[7] adjusted to fit many observationally determined characteristics of SN1993J. Following these authors, we consider the ejecta-density profile $\rho_{ej} = \rho_o(r_o/r)^n(t/t_o)^{(n-3n-3)}$ with reference density $\rho_o$ at radius $r_o$ and time $t_o$, and with $n \sim 30$. It is noteworthy that the ejecta density for SN1993J has a steep density gradient compared with other Type II supernovae. The CSM-density profile is $\rho_{CSM} = \rho_o(r_o/r)^s$, with $s = 1.7$. As the shock wave passes from the steep profile of ejecta to the much-shallower profile in the CSM, a thin, interface-like region forms inside, which the density decreases with radius by a few hundreds. The interface decelerates at a rate of $\sim 25$ cm s$^{-2}$, at $t = 0.1$ years.

The pressure variations are small across this interface region and by the time of SNR formation the radiation pressure is negligible; thus, the temperature drops by a factor of a few hundreds across a distance of a few collisional mean free paths. In response, an intense energy flux forms across the boundary, which will then drive heat into the shocked ejecta. It appears to us that prior simulations have not included this effect. Balancing the energy fluxes, for a typical electron temperature in the shocked CSM of just above $10^9$ K, will sustain a temperature in the ejecta of about $2.5 \times 10^7$ K. This is substantially larger than the temperature ($\sim 4 \times 10^6$ K) produced by the reverse shock in the standard models without heat conduction and will help resist the density increase that radiative cooling of the shocked ejecta would otherwise produce. The increased pressure in the dense material, created by the deposited heat, will cause expansion at the interface, in effect removing layers of RT-unstable material. The characteristic speed for this process is the sound speed, which we estimate as 700 km s$^{-1}$ at 0.1 years.

High energy fluxes affect the emission from an SNR and we hypothesize they might also alter the behavior of the RT. When

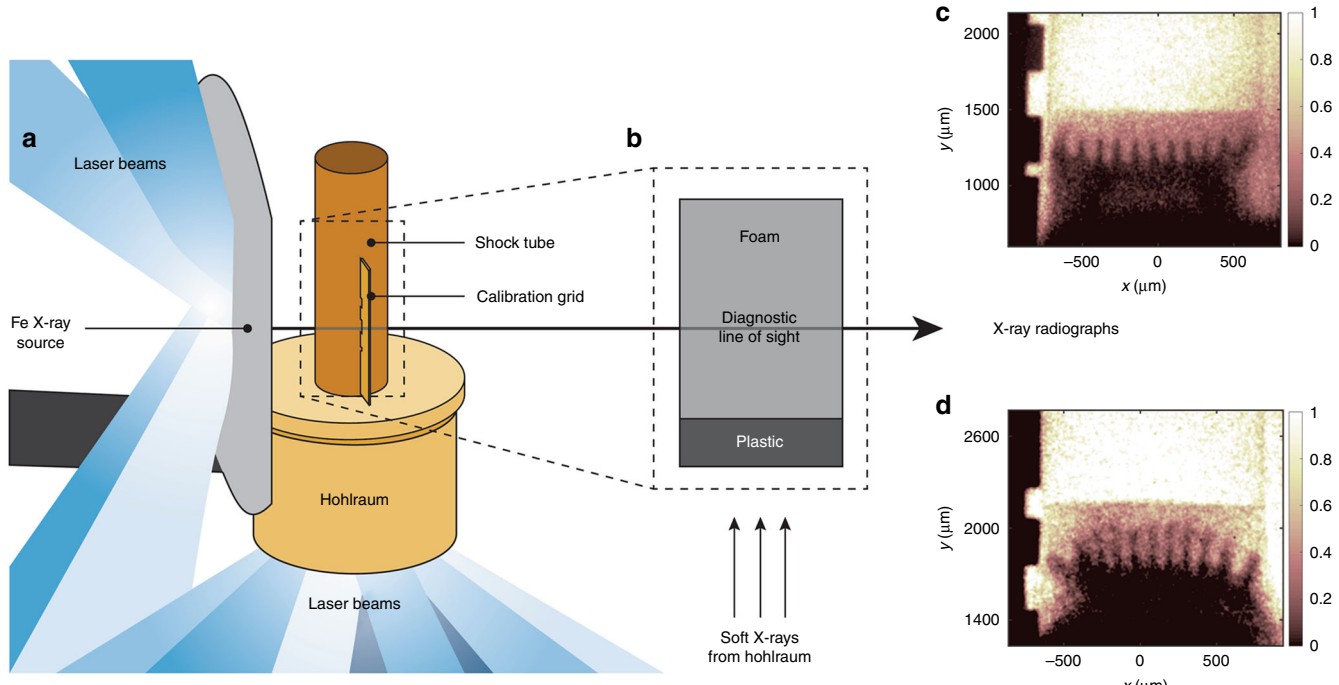

**Fig. 1** Experimental target and radiographs. **a** NIF target schematic with laser beams incident on the gold hohlraum to create the X-ray drive and on the large-area backlighter to create the diagnostic X-ray source. Attached to the hohlraum is a plastic shock tube. The soft X-rays from the hohlraum create a shock wave in the plastic layer inside the shock tube **b**, which decays into a blast wave before crossing the unstable interface and entering the foam. The diagnostic X-ray source creates radiographs by being preferentially absorbed by a tracer layer in center of the plastic. **c**, **d** X-ray radiographs of the experiment. Here, the plasma flows upward and the dark fingers are due to RT instability growth. The color bar indicates the relative transmission for **c** the high-flux case at $t = 13$ ns and **d** the low-flux case taken at $t = 34$ ns. The two experiments have similar RT growth factors, as described in the text

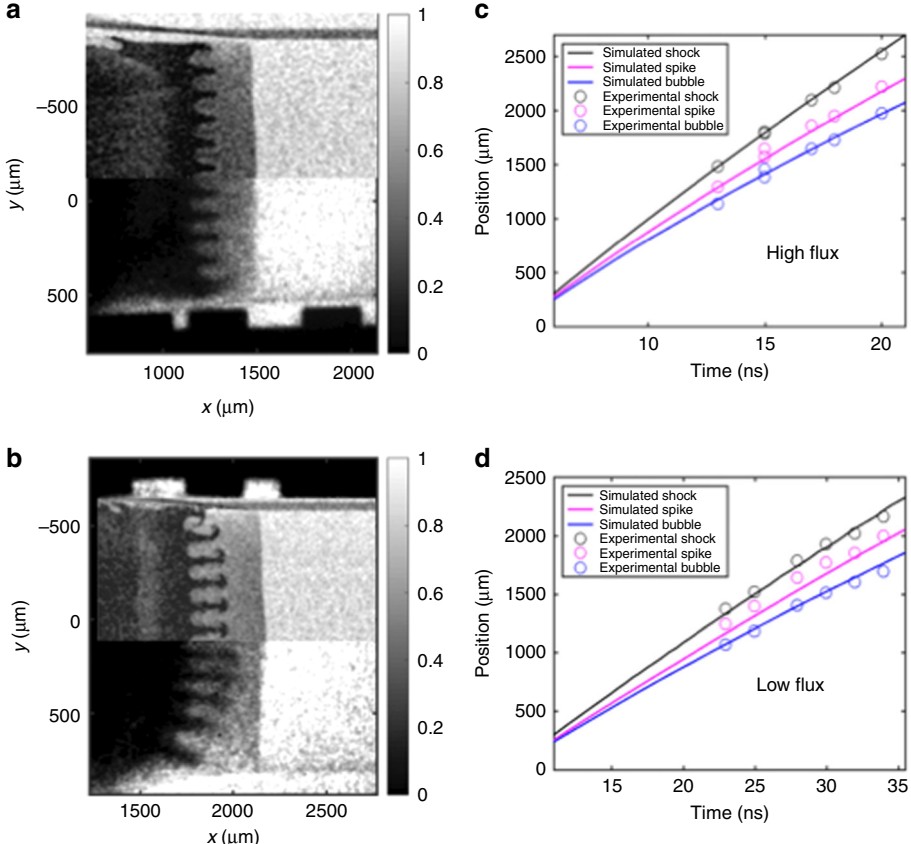

**Fig. 2** Experimental data and results of simulations. In order for the simulation results to reproduce the shock positions, the energy flux driving the dense plastic was reduced to 78% of its nominal value. The simulated and observed radiographs are shown for **a** the high-flux and **b** the low-flux cases. The simulated radiographs are post-processed results from CRASH simulations. **c** and **d** show the feature position vs time for the high-flux and low-flux cases, respectively. Shock, spike, and bubble positions for the simulation results are shown with solid lines and the experimental data points have an error roughly the size of the data point. The simulations were performed in Cartesian ($x$–$y$) geometry

the radiation reaches the outer, RT-unstable surface, we hypothesized that this might have the effect of peeling away some structure and thus changing the initial state for later evolution of RT. We devised an experiment for the National Ignition Facility (NIF)[8] that could produce and allow observations of such an effect. We discovered the apparent, larger role of heat conduction when we closely examined the comparison between the experimental results and the SNR observations and models.

## Results

**Experimental method**. The experiment used NIF to create a hydrodynamically unstable interface subject to a high energy flux by the emergence of a blast wave into lower-density matter, in analogy to the SNR. The following studies discuss the design and development of the experiment in more detail[9–12]. The NIF laser beams irradiate the gold hohlraum, as shown in Fig. 1a, creating X-rays whose absorption creates a pressure impulse by ablation. This launches a blast wave into the package attached to the hohlraum, whose important elements are a plastic layer at 1.4 g cm$^{-3}$ followed by an SiO$_2$ foam that has an initial density of 0.02 g cm$^{-3}$ (see Fig. 1b). When the blast wave crosses the interface, an initial 2D modulation, with a wavelength of 120 μm and an amplitude of 6 μm, is subjected to vorticity generation and hydrodynamic expansion, after which the RT instability produces unstable growth of the structure. High energy fluxes into the

interface can affect RT and the strength of these fluxes depends strongly on the ablation pressure. By varying the hohlraum temperature, we observed the RT growth when the energy fluxes were either important or negligible. We refer to these two cases as high flux and low flux, respectively. Attached to the target is a large Fe foil that produces a bright X-ray source when irradiated with additional laser beams. This source is used to create X-ray radiographs (see Fig. 1c, d), giving us a 2D image of the structure produced by instability growth[13]. Additional target and diagnostic details can be found in Supplementary Note 1.

Figure 1c, d show typical X-ray radiographs from both the high-flux and low-flux experiments. The flow is moving in the upward direction into the unshocked foam at the top of each image. The shock front is at around 1500 and 2000 μm in the high-flux and low-flux radiographs, respectively. Behind the shock is the RT-unstable interface with the dark spikes of dense material alternating with bubbles of lighter material in between. The shape of the spikes and the overall RT growth are different in the two cases, as discussed below.

Multiple X-ray radiographs obtained at different times show the evolution of the instability growth for both cases. In order to meaningfully compare the RT growth of each system, one must account for the different, time-dependent acceleration and Atwood numbers for the two cases. The instantaneous RT growth rate is defined as the exponentiation rate for small-amplitude modulations, $\gamma_{\mathrm{RT}}(t) = \sqrt{A(t) * g(t) * k}$ with units of s$^{-1}$, where $A(t)$ is the time-dependent Atwood number, $A = \frac{\rho_1 - \rho_2}{\rho_1 + \rho_2}$,

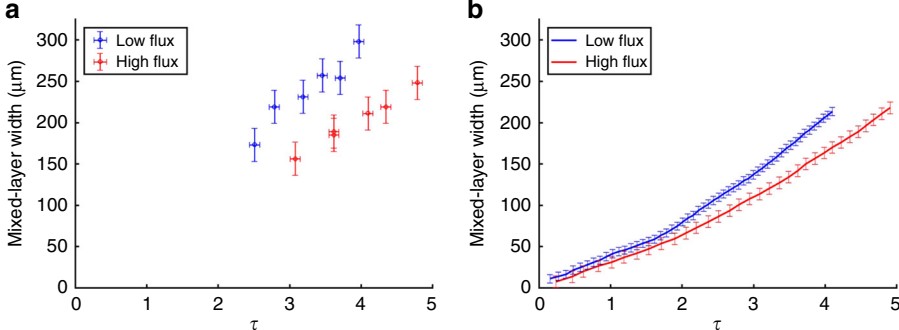

**Fig. 3** Rayleigh–Taylor growth comparison. **a** Experimental mixed-layer width vs the RT growth factor $\tau$, the number of instability e-foldings, $\tau = \int \gamma_{RT}\,dt$ where $\gamma_{RT}(t) = \sqrt{A(t) * g(t) * k}$ for high- and low-flux environments. The mixed-layer is smaller in the high-flux case, indicating a reduction in RT growth. The vertical error bars reflect the uncertainty of the location of the RT spike tip and bubble head, and the horizontal error reflects the uncertainty in the experimental time. **b** Simulation results of mixed-layer width vs $\tau$. The error bars reflect different methods for determining the location of the RT features. These methods include the material interface location and the position determined from simulated radiographs shown in Fig. 2

**Table 1 Dimensionless parameters and their physical meaning**

| Dimensionless number | SN1993J | NIF experiment | Physical meaning |
|---|---|---|---|
| $\lambda_c/L$ | $\sim 10^{-4}$ | $\sim 10^{-8}$ | Highly collisional |
| Re | $\sim 10^{6}$ | $\sim 10^{7}$ | Negligible viscosity |
| Energy flux ratio $R$ | $\sim 10^{3}$ | $\sim 2$ | Energy fluxes are important |

Both SN1993J (at 0.1 years) and the laboratory experiment have characteristic length $L \gg \lambda_c$, the mean free path for ion–ion collisions, in their denser shocked layers. They also have large Reynolds number, $Re = UL/\nu$, where $U$ is the characteristic velocity and $\nu$ is the kinematic viscosity. The text discusses the energy flux ratio $R$

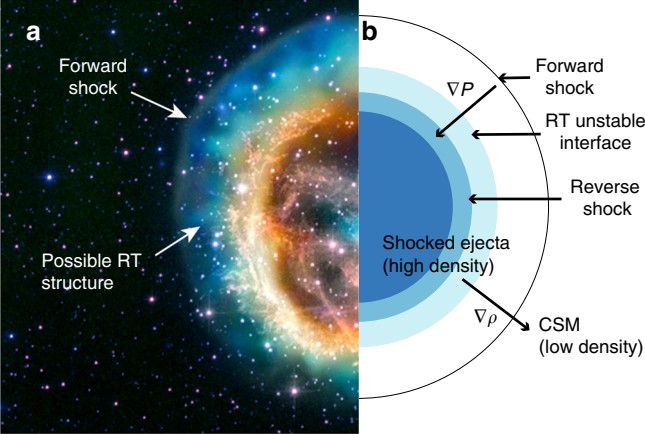

**Fig. 4** Image of supernova remnant. **a** False-color image of SNR E0102.2-72. This object is believed to result from a core-collapse supernova about 1000 years ago. One can see the edge of the forward shock. The modulated boundary within it might be structuring of the ejecta-CSM interface produced by RT. The brighter, inner colors are attributed to emission from the higher-Z, interior portions of the ejecta. We credit John Hughes of Rutgers University with having called the potential connection to RT to our attention. Image credit: X-ray (NASA/CXC/MIT/D. Dewey et al. and NASA/CXC/SAO/J. DePasquale); Optical (NASA/STScI). **b** Schematic (size and shape not to scale) of inner structures of the supernova that creates the opposing density and pressure gradients to create an RT unstable interface

in which $\rho_1$ is the heavy fluid and $\rho_2$ is the lighter fluid. The time-dependent acceleration is given by $g(t)$ and $k$ is the wave number of the initial seed perturbation. The non-dimensional RT growth factor is $\tau = \int \gamma_{RT}\,dt$, which is evaluated after RT growth begins. The growth factor used here does not include ablation or other stabilizing effects.

Simulations performed using the multidimensional radiation hydrodynamics code CRASH[14–16], adjusted to match the one-dimensional dynamics of the experiment, provided values of $A(t)$ and $g(t)$. Figure 2a, b show a comparison of the simulated and experimental radiographs for both cases. The simulated radiographs include realistic levels of Poisson noise and smearing over the resolution element of the diagnostic. Figure 2c, d compare the simulated and the observed positions of relevant flow structures. The simulation accurately predicts the spike- and bubble-tip locations for the high-flux case, but appears to produce shorter spikes than are observed for the low-flux case. This is not immediately understood, but may be related to similar observations in a previous case[17].

**RT instability**. The high-flux and low-flux cases show distinct differences in both simulations and experiments; the overall appearance of the spike is different, with the characteristic mushroom cap for RT spikes not being present in the high-flux case. The mixed-layer width is the difference between the spike- and bubble-tip locations. Figure 3a shows that the experimental mixed-layer width is larger for the low-flux case than for the high-flux case for any given non-dimensional growth factor $\tau$. Figure 3b shows mixed-layer width for high- and low-flux cases extracted from CRASH simulation results. This indicates that the radiation has affected the growth of the RT instability in a similar way, in both the experiment and simulation.

**Discussion**
Appropriate dimensionless parameters inform the comparison of disparate systems and suggest directions for further analysis. Table 1 shows three of these. Both the NIF experiment and the SNR that it aims to describe have collisional ions and so can be thought of as fluids. Both have large Reynolds number, Re, and so viscosity is not central to their evolution. The parameter $R$ is the ratio of the combined radiative and conductive energy fluxes

incident toward the interface to the mechanical energy flux, $F_{mech}$, which sustains the reverse shock. $R$ is > 1 for the experiment so that one expects the energy flux to have important effects, as observed. We estimate $R$ to be very large in the SNR. These energy fluxes are discussed in detail in the Supplementary Note 3. In the case of the inner interface surface, radiative fluxes from the shocked ejecta toward the interface are ~ 100 times $F_{mech}$ at 1 year and the ratio declines with time.

RT is stabilized for wavelengths shorter than a limit whose order of magnitude is $2\pi v_a^2/g$, in which $v_a$ is the speed at which material is removed from the interface by ablation or expansion. In the model of SN1993J, the expansion through the interface is sufficient to stabilize RT at all spatial scales up to ~ $0.2r_c$, where $r_c$ is the interface position, and does so independently of time. In the laboratory case, taking $v_a$ from simulations[9], wavelengths below a value of order 180 μm are stabilized. The exact value depends on the precise details of the stabilization mechanism. See Supplementary Note 3 for more details. The observed structures in the experiment might be unstable at a reduced growth rate, or might reflect the inability of the radiation to completely ablate the structures resulting from the initial deposition of vorticity and expansion. There are also some suggestive astrophysical data from SNR E0102.2-72[18], shown in Fig. 4. The modulations seen within the forward shock have a long-wavelength structure and are reasonably consistent in size with the estimate just given. The right side of Fig. 4 shows a schematic of the structure inside the supernova that gives rise to the RT instability. Additional details about the structure of the star is given in Supplementary Note 3.

Our NIF experiment showed that the growth of structure at an unstable interface can be affected by the presence of a high energy flux onto that interface, by comparison of results with low or high fluxes at hydrodynamically equivalent times. Observations of SN1993J motivated the experiment. Examination of the SNR dynamics shortly after the shock breakout led us to discover that the conductive heat fluxes from the shocked CSM into the shocked ejecta are much larger and also more enduring than radiative fluxes. Our analysis is generally applicable and implies that these heat fluxes may have substantial consequences for the structure and stability of the young SNRs. In the case of SNR E0102.2-72, observations seem consistent with our analysis. We conclude that realistic models of SNRs must account for the effects of thermal conduction to accurately predict their evolution at epochs immediately following the shock breakout.

## Methods

**Experiments**. The experiment was performed at the NIF where 64 laser beams irradiated the inner wall of a hohlraum, a gold cylinder having an opening for the laser beams on the bottom and an opening on the top through which soft X-rays irradiated the dense plastic. The hohlraum serves to convert the laser energy to soft X-rays with high efficiency, producing a quasi-Planckian spectrum with a temperature of some hundreds of electronvolts (a few million kelvin) incident on the target surface. An image of the target is shown in Supplementary Fig. 1 and a further description of the experimental conditions is discussed in Supplementary Note 1.

**X-ray radiography**. X-ray radiography captured images of the target, enabling us to observe the unstable interface. Twenty-eight beams illuminated a spot on an iron foil, ~ 2 mm in diameter. The hot iron plasma generated characteristic K-shell X-rays from the He-like, $24^+$ state at ~ 6.7 keV, which traversed the shock tube and were collected using a time-gated X-ray detector. A pinhole with a diameter of 25 μm was set 131 mm from the target to resolve the X-ray signal onto the detector plane, which was 783 mm from the pinhole and produced an image magnified six times. Time-gating was provided by a two-strip microchannel plate coupled to a scintillator screen and the signal was collected on X-ray film. The effective duration of the X-ray pulse was ~ 2.5 ns; thus, the pair of images taken on a single shot were separated by only 2 ns and several shots were necessary to observe the long-term evolution of structure at the interface.

**Numerical simulations**. CRASH is an Eulerian code, which uses tabular equations of state and has dynamic, adaptive-mesh-refinement. Energy transport includes a flux-limited, multigroup-diffusion treatment of radiation and flux-limited electron heat conduction. We used the time-dependent soft-X-ray energy flux to drive the experimental package in a 2D simulation. We processed the simulation results to create simulated radiographs. Further detail about the simulations are provided in Supplementary Note 2.

**Data availability**. The data discussed in this paper are available from the authors on request.

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

## Acknowledgements

We thank NIF Operations and Lawrence Livermore National Laboratory Target Fabrication Facility. This work is supported by the NNSA-DS and SC-OFES Joint Program in High-Energy-Density Laboratory Plasmas under grant number DE-NA0002956 and under the auspices of the U.S. Department of Energy by Lawrence Livermore National Laboratory under Contract DE-AC52-07NA27344 and Los Alamos National Laboratory under U.S. DOE-NNSA Contract DE-AC5206NA25396. We also acknowledge the Lawrence Livermore National Laboratory High Energy Density Summer Student Program.

## Author contributions

C.C.K., H.-S.P., A.R.M., B.A.R., H.F.R., and R.P.D. initially conceived this work. C.C.K., H.-S.P., C.M.H., J.K., and K.A.F. led the experiments and developed the experimental platform. C.C.K., H.-S.P., C.M.H., and W.C.W. analyzed the data. C.C.K., T.P., T.A.H., and R.P.D. developed the astrophysical relevance of this work. C.M.H., A.R.M., H.F.R.,

K.R., S.M., F.W.D., and S.P. performed design simulations in support of platform development. T.P., M.R.T., D.S., A.S., G.M., T.A.H., and M.J.G. performed additional simulations and contributed to data interpretation. C.M.K., S.R.K., E.C.H., R.W., D.C.M., D.K., and E.G. contributed to the support of the experiments. C.C.K., C.M.H., and R.P.D. wrote the paper.

## Additional information

**Competing interests:** The authors declare no competing interests.

