## [Peer Review File · Nature Communications]

Reviewers' comments:

Reviewer #1 (Remarks to the Author):

This paper proposes an interesting joint analysis of a series of laboratory experiments and numerical simulations to investigate the role of the heat transfer in the convective instabilities produced during the early evolution of shock waves of supernova remnants. It is proposed that heat transfer can modify the hydrodynamic instabilities at the interface between the shocked circumstellar material and the shocked ejecta. This effect is claimed to have been mistakenly overlooked in the previous literature. I would recommend a revision of the present form of the paper.

1) I could not find evidence in the paper that the shocked layer in the SNR is collisional that is essential for the heat transfer believed by the authors responsible of stabilising RTIs in the high flux case. Using the reference density of Suzuki et al. 1995 ρ_0 reported in this paper and the post-shock temperature in Eq. 9, the Coulomb collision mean free path would be of the order 10^{15} cm, that does seem comparable or greater than the size of the system, hence collisionless.

2) The RTIs evolve differently in 2 or 3 space dimensions. Could the authors provide evidence that the RTs structures found in the experiment are comparable with a 3D simulation?

3) I am not sure that the relevance of the heat flux from the shocked CSM into shocked ejecta is of general relevance for realistic models of SNR early evolution. The ejecta density in 1993J is exceptionally steep with respect to standard Type IIs (this is not even mentioned in the abstract) with an exceptionally high temperature gradient within the shocked layer. For most Type IIs the heat flux might be not dominant.

4) If the acceleration of the interface $g(t)$ in the experiment is constant, the protruding of RT structures into shocked CSM can be fit with a second order polynomial (Dwarkadas, 541, 418, 2000, Fig. 5). If the time-dependence of $g(t)$ is self-similar, the protruding can be calculated analytically (see Fraschetti et al., 5115, A104, 2010, Eq. 33). What time-dependence are used in the simulations for $A(t)$ and $g(t)$?

5) I would strongly recommend a more careful editing: there are undefined quantities (e.g. ν_{ee} at page 6 of Supp. Material, 7 lines below Eq. 12), temperature in Eq. 9 is not defined, only scaling is provided. Is 27 the exponent of χ in Eq. 7 ?

Reviewer #2 (Remarks to the Author):

This paper deals with Rayleigh-Taylor instabilities (RTIs) in laser experiments and young supernova remnants (SNRs). Particular attention is given to the role of energy fluxes, including heat conduction. The results should be of interest to those working in the laser and SNR communities. However, the results were not clearly presented and the authors missed some points in the previous literature. My comments are as follows:

1. The authors discuss 3 energy fluxes that might affect the RTI region: the cooling radiation from the reverse shock front (as probably occurs in SN 1993J), the radiative flux from the hohlraum (as in the experiment and the simulation of the experiment), and the heat conduction flux. The importance of heat conduction is deduced from approximate arguments and not detailed simulations, although it is stated that the code CRASH includes flux-limited electron heat transport. The authors suggest that a

heat conduction flux and a radiative flux passing through the unstable region have similar effects, but the paper would be improved if they carried out a simulation with heat conduction.

2. There have been discussions of heat conduction for the SN case in 1-dimension, although the problem has not been solved. I would note Bedogni & D'Ercole 1988, A&A, 190, 320 and Band, D. 1988, ApJ, 332, 842, but do not know of multi-dimensional simulations.

3. In a case like SN 1993J, the standard view is that there is cooling reverse shock wave at early times that is able to absorb and reradiate the emission from immediately behind the shock. The gas is thick to photoelectric absorption. The RTI for the cooling reverse shock case is discussed by Chevalier & Blondin 1995, ApJ, 444, 312. The authors here invoke photoelectric heating by the cooling radiation, but it seems that would be important only during the transition from radiative to non-radiative reverse shock. This is stated on p. 7, but not on p. 5 when SN 1993J is initially discussed.

4. p. 7. The authors combine radiative and conductive fluxes in the R parameter. It would be clearer to keep them separate; the fluxes may not operate in exactly the same way.

5. In their discussion of SN 1993J, the authors refer to the paper Fransson et al. 1996 for the basic parameters. That paper was incorrect in neglecting synchrotron self absorption for the radio emission. A better model, with different parameters, is described in Fransson & Bjornsson 1998, ApJ, 509, 861.

6. When the authors discuss the possible magnetic inhibition of conduction (p. 6 of SM), it is worth noting that quite high B fields are inferred from synchrotron radiation from the supernova interaction region (e.g. Fransson & Bjornsson 98).

The paper should be suitable for publication after these points are addressed.

Reviewer #3 (Remarks to the Author):

The Kuranz's article is based on two experimental observations and the main point is the strong influence of the electron heat conduction in the Rayleigh-Taylor (RT) instability evolution, higher than radiation or mechanical energy transfer. The first experiment is a laboratory experiment performed at NIF facility. The target used in the experiment is well characterized in terms of dimensions, materials and illumination conditions, in this case with a half hohlraum. The target is similar to other used for RT studies, and close to the one performed six years ago at Omega laser facility by the same first author. All the diagnostic used is well known and used in many previous successful experiments. The numerical simulations with the 2D AMR radiation diffusion CRASH code gives a close agreement with the experiments in terms of bubble-spike position. The theoretical analysis of the lab experiment is lacking, but is well known from a lot of previous RT and related experiments, and supported by numerical codes that includes all the relevant physics. The fig 3 is the central part of the reasoning and supports the idea of the stabilizing effect of radiation and electron flux, even for a ratio R of 2. In this figure is missing more points for high and low fluxes for the same tau parameter, and this is a weak point in the reasoning. Fig 3 of suppl. material is the numerical simulation with low flux (low opacity) and high flux. I really miss the results of the simulations of the experiments compared with the experimental data drawn in fig 3. It will do a much stronger reasoning. In any case the experimental results are clear and the idea of heat flux (rad plus elec) modifying strongly the RT growth is well proved.

The main issue with the article is related to the astrophysical side. As far as I know, there are no simulations of the early SNR evolution with electron and radiation energy transport, but codes as FLASH or CASTOR are able to do the work because they have implicit conduction (in FLASH) and radiative transport. The argument of the importance of electron energy flux in the early stage of SNR evolution will be stronger if the authors test this hypothesis with simulations. The analytic work devised at the supplement material makes clear that the high temperature difference will produce a huge Q_e , even limited to a 10% of the free streaming value. The comparison with SN 1993J is meaningful and reinforces the main conclusion of the article. But the comparison with the E0102.2 is very speculative and based only on a visual comparison, as this SNR is probably very asymmetrical and the view is from an expansion line.

Another issue is about scaling between astro and lab cases. As the authors point out, the heat (rad+elec) fluxes are important in both astro and lab cases, but the scaling for this case is more complicated and analysis based in Ryutov (199), Falize (2011) and Cross (2014) would be necessary. Dimensionless numbers are mandatory.

Briefing, the article is very appropriate for the nat comm journal since it compels the scientific community to rework the calculations of the evolution of RT in supernovae type II remnants taking into account the electron heat conduction. Previously it was assumed that for the evolution of early SNR the main energy transport mechanism is mechanical and radiation. But the article needs a revision in two parts

- a) To compare the simulations with astrophysical ones with electron and radiative energy fluxes
- b) To show explicitly the scaling between astro and lab cases

Besides this there are several comments to the text

1. Is the tau parameter (pg 3) standard in RT studies?
2. Fig 3 should be superimposed with numerical simulations results
3. r_c in page 7 is not defined (is the interface position, I guess)
4. In Fig4. (right) is supposed a spherical expansion, but experimental data suggest an asymmetrical one. It should be written to clarify the comparison.
5. Fig 3 of suppl material, the axis Y should give the actual dimension, as length dimensions appear at the X axis.
6. Fig 4 should have plotted the temperature profile too, as is essential in the reasoning of the article.
9. The table 1 of both article and suppl material is missing the main dimensionless parameters as Peclet number, Radiation number, Euler number.

So I recommend the publication of the article with the corrections mentioned above.

Reviewer #1 (Remarks to the Author):

This paper proposes an interesting joint analysis of a series of laboratory experiments and numerical simulations to investigate the role of the heat transfer in the convective instabilities produced during the early evolution of shock waves of supernova remnants. It is proposed that heat transfer can modify the hydrodynamic instabilities at the interface between the shocked circumstellar material and the shocked ejecta. This effect is claimed to have been mistakenly overlooked in the previous literature. I would recommend a revision of the present form of the paper.

1) I could not find evidence in the paper that the shocked layer in the SNR is collisional that is essential for the heat transfer believed by the authors responsible of stabilising RTIs in the high flux case. Using the reference density of Suzuki et al. 1995 ρ_0 reported in this paper and the post-shock temperature in Eq. 9, the Coulomb collision mean free path would be of the order 10^{15} cm, that does seem comparable or greater than the size of the system, hence collisionless.

We concur with the referee that the shocked layer is collisionless (or nearly so). We did a poor job of communicating the fact that we used the correct heat-flux model for this case. We have revised the text in the supplemental material to clarify this point. The text now states:

“In many contexts, heat conduction is a diffusive process. However this requires that the temperature scale length is quite large ($\gg 30$ mean-free-paths for the high-temperature particles¹⁵). In contrast, in the regime of interest here, the collisional mean-free-path is larger than the temperature scale lengths seen in the referenced calculations. This is a common regime for laser-produced plasmas and other strongly-heated laboratory plasmas. In such cases, the electron heat flux Q_e can be approximated as a fraction, of order 10%, of the free-streaming heat flux, so that..”

2) The RTIs evolve differently in 2 or 3 space dimensions. Could the authors provide evidence that the RTs structures found in the experiment are comparable with a 3D simulation?

We concur that RTI evolves differently in 2D and 3D. We did not clearly express that the modulation in the experiment is 2D. We have edited the text as follows

“When the blast wave crosses the interface, an initial 2D modulation, with a wavelength of 120- μ m and an amplitude of 6- μ m, is subjected to vorticity generation and hydrodynamic expansion, after which the RT instability produces unstable growth of the structure.”

3) I am not sure that the relevance of the heat flux from the shocked CSM into shocked ejecta is of general relevance for realistic models of SNR early evolution. The ejecta density in 1993J is exceptionally steep with respect to standard Type IIs (this is not even mentioned in the abstract) with an exceptionally high temperature gradient within the shocked layer. For most Type IIs the heat flux might be not dominant.

We have edited the abstract to mention the steepness of the ejecta in SN1993J:

In analyzing the comparison with SN 1993J, a Type II supernovae, where the ejecta density has a steep density gradient compared to other Type II supernovae, we discovered...”

The question of how the impact of the heat flux might change for SNRs of varying steepness, raised by the referee, is certainly relevant. To address this, we have added a sentence to the discussion in the supplemental material. The relevant portion now reads:

For SN 1993J, the magnitude of R is ~ 103 , it is dominated by heat conduction, based on the calculations above, and it is independent of time (subject to the continuing validity of the specific models used). For less steep SNRs (having smaller values of n), R decreases, but only a factor of about ten as one progresses from $n = 30$ to $n = 10$. As a result, the qualitative effect identified here seems likely to remain.

4) If the acceleration of the interface $g(t)$ in the experiment is constant, the protruding of RT structures into shocked CSM can be fit with a second order polynomial (Dworkadas, 541, 418, 2000, Fig. 5). If the time-dependence of $g(t)$ is self-similar, the protruding can be calculated analytically (see Frascchetti et al., 5115, A104, 2010, Eq. 33). What time-dependence are used in the simulations for $A(t)$ and $g(t)$?

$A(t)$ and $g(t)$ are ascertained from the CRASH simulations and not from a self-similar model. This detail, unfortunately, was left out of the paper. We have included the following discussion about the acceleration and Atwood number in the Supplemental Material.

The interface acceleration, $g(t)$, is the time derivative of the velocity of the interface, which is determined from a simulation without a seeded perturbation (ie flat plastic-foam interface). For the high-flux case $g(t) = -18.8t^{-0.8}$ and in the low-flux case $g(t) = -11.5t^{-0.9}$. In both cases, g is in $\mu\text{m/ns}^2$ and t is in ns. For the Atwood number the material densities are taken at the local maximum density and local minimum density near the interface position ignoring features that are caused by material property discontinuities. In both cases, the post-shock Atwood number is taken relatively constant and is 0.5 and 0.7 in the high-drive and low-drive case, respectively.

5) I would strongly recommend a more careful editing: there are undefined quantities (e.g. ν_{ee} at page 6 of Supp. Material, 7 lines below Eq. 12), temperature in Eq. 9 is not defined, only scaling is provided. Is 27 the exponent of χ in Eq. 7 ?

We defined ν_{ee} in the line below where it is used and we have included an additional equation that defines T in general. The exponent of χ_r is $1/(n-3)$, which is 27 for $n = 30$, but n was substituted too early. The equation has been edited.

Reviewer #2 (Remarks to the Author):

This paper deals with Rayleigh-Taylor instabilities (RTIs) in laser experiments and young supernova remnants (SNRs). Particular attention is given to the role of energy fluxes, including heat conduction. The results should be of interest to those working in the laser and SNR communities. However, the results were not clearly presented and the authors missed some points in the previous literature. My comments are as follows:

1. The authors discuss 3 energy fluxes that might affect the RTI region: the cooling radiation from the reverse shock front (as probably occurs in SN 1993J), the radiative flux from the hohlraum (as in the experiment and the simulation of the experiment), and the heat conduction flux. The importance of heat conduction is deduced from approximate arguments and not detailed simulations, although it is stated that the code CRASH includes flux-limited electron heat transport. The authors suggest that a heat conduction flux and a radiative flux passing through the unstable region have similar effects, but the paper would be improved if they carried out a simulation with heat conduction.

Our simulations did include heat conduction, but we failed to make this clear in the text. It turns out, as we show in Table 2 in the Supplemental Material, that diffusive energy transport dominates in the laboratory case while energy transport by electron heat conduction dominates in the SNR. Since both add to the thermal energy of the matter, we believe that their physical effects are comparable.

We have edited the text regarding the code to read:

CRASH is an Eulerian code, which uses tabular equations of state and has dynamic, adaptive-mesh-refinement. Energy transport includes a flux-limited, multigroup-diffusion treatment of radiation and flux-limited electron heat conduction,

2. There have been discussions of heat conduction for the SN case in 1-dimension, although the problem has not been solved. I would note Bedogni & D'Ercole 1988, A&A, 190, 320 and Band, D. 1988, ApJ, 332, 842, but do not know of multi-dimensional simulations.

We have added the following text:

There have been discussions of heat conduction for the SN case in 1-dimension, although the problem has not been solved \citep{Bedogni, Band} We are not aware of any multi-dimensional simulations including these effects.

3. In a case like SN 1993J, the standard view is that there is cooling reverse shock wave at early times that is able to absorb and reradiate the emission from immediately behind the shock. The gas is thick to photoelectric absorption. The RTI for the cooling reverse shock case is discussed by Chevalier & Blondin 1995, ApJ, 444, 312. The authors here invoke photoelectric heating by the cooling radiation, but it seems that would be important only during the transition from radiative to non-radiative reverse shock. This is stated on p. 7, but not on p. 5 when SN 1993J is initially discussed.

The following material has been moved to earlier in the paper, when SN1993J is initially discussed.

This radiation is incident from within the shocked ejecta and so will only affect the interface during the phase when the dense layer at the interface becomes partially transparent. In contrast, the conductive fluxes from the CSM through the interface are present continually. They are large enough that they might fundamentally change the commonly assumed structure of this region. This motivates future experimental and theoretical studies.

4. p. 7. The authors combine radiative and conductive fluxes in the R parameter. It would be clearer to keep them separate; the fluxes may not operate in exactly the same way.

In the main text we have added the text below, directing the reader to the SM where the energy fluxes are discussed both separately and in more detail.

These energy fluxes are discussed in detail in the Supplemental Material.

5. In their discussion of SN 1993J, the authors refer to the paper Fransson et al. 1996 for the basic parameters. That paper was incorrect in neglecting synchrotron self absorption for the radio emission. A better model, with different parameters, is described in Fransson & Bjornsson 1998, ApJ, 509, 861.

We have added text to the discussion in order to cite the paper mentioned and clarify what we did, as follows:

We note that Fransson et al \cite{Fransson:ApJ98} prefer $s = 2$. We use the model of \cite{Suzuki:ApJ95:93J} below, but note that the conclusions drawn are not sensitive to the exact choice of parameters, and also would apply to the cases having much shallower ejecta profiles that are thought to be more typical.

6. When the authors discuss the possible magnetic inhibition of conduction (p. 6 of SM), it is worth noting that quite high B fields are inferred from synchrotron radiation from the supernova interaction region (e.g. Fransson & Bjornsson 98).

Contemplating the field as evaluated by Fransson & Bjornsson led us to revise the discussion of heat conduction in the magnetized case. But, as indicated previously, this does not change the order of magnitude of the anticipated heat flux. We replaced the previous paragraph about this subject with the following:

As it turns out the model just discussed is also relevant to the magnetized, turbulent state that is inferred to exist in SNR 1993J. From an analysis of the measured radio spectra, attributed to synchrotron emission \cite{Fransson:ApJ98} infer that the magnetic field in the shocked CSM is probably tens of Gauss during the time period when energy-flux effects might be significant. This represents a significant fraction (their number is 14%) of the local thermal energy. For this high a magnetic field, the electrons would be very well

magnetized. The condition for the electrons to be well-magnetized is $\omega_{ce}/\nu_{ee} = 1$, in which $\omega_{ce} = eB/(m_e c)$, for Gaussian cgs units with electric charge e , magnetic field B , and light speed c , and $\nu_{ee} = 4.2 \times 10^{-6} n_e \log \Lambda / T_{eV}^{3/2}$, with electron temperature in eV being T_{eV} and Coulomb logarithm being $\log \Lambda$. One finds the magnetic field required for magnetization to matter to be $\sim 100 \mu\text{Gauss} \sim 10 \text{ nT}$ in the SNR and $\sim 100 \text{ MGauss}$ in the lab experiment. The field is far smaller than 100 MGauss in the lab, but far larger than 100 μGauss in the SNR.

It turns out that, in such a magnetized case, the heat transport is also reasonably estimated by Eq.~\ref{eq:Qe}. Ryutov et al. \cite{Ryutov:ApJ99} discuss heat transport in a generic, magnetized plasma. Assuming the magnetic field within the CSM to be turbulent, as seems likely and is the case in the Solar wind, they find a minimum value for the kinematic coefficient of heat conduction to be $\nu_{e, cs}$, in which the ion Larmor radius of gyration in the magnetic field is r_{Li} . If the scale height of the temperature change is $10 r_{Li}$, then this implies a heat flux of order that of Eq.~\ref{eq:Qe}. If the temperature gradient decreases, the heat flux would go down, but by then the surface of the dense ejecta will be heated and will ablate, influencing the RT instability. A simpler way to make the same point is that in the turbulent- (Bohm-) diffusion limit, the electrons leave the hotter matter by moving some fraction $(\sim 1/16)$ of a gyro orbit radius in a time $(1/\omega_{ci})$, where ω_{ci} is the electron cyclotron frequency in radians/s. The ratio of this distance to this time gives a speed $\sim \nu_{e, cs}/16$ and thus a heat flux roughly consistent with Eq.~\ref{eq:Qe}.

The paper should be suitable for publication after these points are addressed.

Reviewer #3:

Reviewer #3 (Remarks to the Author):

The Kuranz's article is based on two experimental observations and the main point is the strong influence of the electron heat conduction in the Rayleigh-Taylor (RT) instability evolution, higher than radiation or mechanical energy transfer. The first experiment is a laboratory experiment performed at NIF facility. The target used in the experiment is well characterized in terms of dimensions, materials and illumination conditions, in this case with a half hohlraum. The target is similar to other used for RT studies, and close to the one performed six years ago at Omega laser facility by the same first author. All the diagnostic used is well known and used in many previous successful experiments. The numerical simulations with the 2D AMR radiation diffusion CRASH code gives a close agreement with the experiments in terms of bubble-spike position. The theoretical analysis of the lab experiment is lacking, but is well known from a lot of previous RT and related experiments, and supported by numerical codes that includes all the relevant physics. The fig 3 is the central part of the reasoning and supports the idea of the stabilizing effect of radiation and electron flux, even for a ratio R of 2. In this figure is missing more points for high and low fluxes for the same tau parameter, and this is a weak point in the reasoning.

Fig 3 of suppl. material is the numerical simulation with low flux (low opacity) and high flux. I really miss the results of the simulations of the experiments compared with the experimental data drawn in fig 3. It will do a much stronger reasoning. In any case the experimental results are clear and the idea of heat flux (rad plus elec) modifying strongly the RT growth is well proved.

The main issue with the article is related to the astrophysical side. As far as i know, there are no simulations of the early SNR evolution with electron and radiation energy transport, but codes as FLASH or CASTOR are able to do the work because they have implicit conduction (in FLASH) and radiative transport. The argument of the importance of electron energy flux in the early stage of SNR evolution will be stronger if the authors test this hypothesis with simulations. The analytic work devised at the supplement material makes clear that the high temperature difference will produce a huge Q_e , even limited to a 10% of the free streaming value. The comparison with SN 1993J is meaningful and reinforce the main conclusion of the article. But the comparison with the E0102.2 is very speculative and based only on a visual comparison, as this SNR is probably very asymmetrical and the view is from a expansion line.

Another issue is about scaling between astro and lab cases. As the authors point out, the heat (rad+elec) fluxes are important in both astro and lab cases, but the scaling for this case is more complicated and analysis based in Ryutov (199), Falize (2011) and Cross (2014) would be necessary. Dimensionless number are mandatory.

Briefing, the article is very appropriate for the nat comm journal since it compels the scientific community to rework the calculations of the evolution of RT in supernovae type II remnants taking into account the electron heat conduction. Previously it was assumed that for the evolution of early SNR the main energy transport mechanism is mechanical and radiation. But the article needs a revision in two parts

a) To compare the simulations with astrophysical ones with electron and radiative energy fluxes

We attempted, using a version of the FLASH code and a different one-dimensional code, to model the evolution of interest. Because of the enormous range of scales involved in this phase of the evolution from SN to SNR, we proved unable to accomplish this within a few months of postdoc time. We hope that publication of this paper will enable us or someone to find funding to support simulations of this difficult phase of SN/SNR behavior.

Also, a theoretical analysis of the lab experiment is underway and the author and co-authors are preparing a manuscript for submission. We did not feel that it added to the astrophysical case and decided to present it in a separate publication.

b) To show explicitly the scaling between astro and lab cases

We have added more information in the Supplemental material in regards to dimensionless parameters. More details are provided in Remark #9 below.

Besides this there are several comments to the text

1. Is the tau parameter (pg 3) standard in RT studies?

It is common to use nondimensional quantities in fluid dynamics (and hydrodynamic instability studies). The parameter tau is used in other papers, but it not unique to any work so we do not believe that it warrants a citation. We did also include a definition of tau in the Figure 3 caption for further clarity.

2. Fig 3 should be superimposed with numerical simulations results

We have added another figure with numerical simulation results. There are differences in the experimental and simulation results due to simulation tuning and specific solvers used in the code. This is addressed when discussing Figure 3. However, both the experiment and simulation show the same effect regarding the reduction in the RT due to high-energy fluxes.

The text below has been added to support the additional panel in Figure 3.

(Left) Experimental mixed-layer width vs. the RT growth factor $\tau = \int \gamma_{RT} dt$ where $\gamma_{RT}(t) = \sqrt{A(t)g(t)k}$ for high- and low-flux environments. The mixed-layer is smaller in the high-flux case, indicating a reduction in RT growth. (Right) Simulation results of spike and bubble height measured from the interface ($y = 0$).

Also on this point, in the general comments the reviewer noted:

In this figure is missing more points for high and low fluxes for the same tau parameter, and this is a weak point in the reasoning.

We changed the bounds for evaluating tau to after RT growth has begun. We had previously evaluated it at the $t=0$ for the experiment (i.e. when the laser beams irradiated the hohlraum). Our current method is more relevant for the comparison in the manuscript and now there is considerably more overlap in the 2 cases. We have edited the text to reflect this.

The non-dimensional RT growth factor is $\tau = \int \gamma_{RT} dt$, which is evaluated after RT growth begins.

3. rc in page 7 is not defined (is the interface position, I guess)

Yes, r_c is the interface position. We have edited the text as follows

$\sim 0.2 r_c$, where r_c is the interface position

4. In Fig4. (right) is supposed a spherical expansion, but experimental data suggest an

asymmetrical one. It should be written to clarify the comparison.

The image was meant to illustrate the directions of shocks and gradients in a SN and not to indicate the size or shape. We have added the caveat below to the figure caption.

“Schematic (size and shape not to scale) of inner structures...”

5. Fig 3 of suppl material, the axis Y should give the actual dimension, as length dimensions appear at the X axis.

This has been changed.

6. Fig 4 should have plotted the temperature profile too, as is essential in the reasoning of the article.

This has been added to Figure 4.

9. The table 1 of of both article and suppl material is missing the main dimensionless parameters as Peclet number, Radiation number, Euler number.

We have added more information in the Supplemental material in regards to dimensionless parameters.

For the Peclet number in the diffusive regime, we have LU/χ , where χ is the thermal diffusivity. However, for free-streaming heat flow the heat flux is $Q = \frac{3}{2} f p v_{\text{the}}$ with electron thermal speed v_{the} and flux-limiter f , which is about 0.1. Therefore, in the free-streaming regime, we have $Pe_{\text{free}} = 2U/3fv_{\text{the}}$.

To further consider the similarity of the systems we look at the Ryutov number, $Ry = v^(\rho^*/P^*)^{1/2}$ (sometimes called the Euler number). When comparing two systems, most dimensionless numbers do not need to be the same, simply in the same regime. However, for two systems to fulfill the conditions for hydrodynamics similarity, as detailed by Ruytov et al. \cite{Ruytov:ApJ99}, Ry must be the same in both systems. While Ry for both systems is similar (see Table~\ref{tab:dimensionless}), these estimates are based on many assumptions and could vary by at least an order of magnitude. In general, scaling to a specific object is not the main goal of this work, but to show the importance of energy fluxes in the evolution of young supernova remnants.*

Reviewers' comments:

Reviewer #1 (Remarks to the Author):

I would like to thank the authors to carefully revise the paper and answer my questions. However, I would like a clarification about some points, not yet clear in my opinion.

About my previous point 1), I think consistency in definition and terminology would help clarity. Authors report in table 1 that the performed NIF experiment is as much collisional as the SN 1993J. If laboratory experiments are usually considered collisional because of the extremely large electron density, on the other hand astrophysical shocks, and so shocked layers in SN, are typically collisionless: Coulomb collision mean free path is much smaller than the typical scale of the system. Table 1 gives a very small λ_c (that is not defined but is presumably such mean free path). In their reply, the authors define the shocked layer collisionless because the mean free path is much larger than the temperature scale height, whose definition I could not find. So a consistency would be appreciated.

The authors are merging the language of two separate communities, one comparing L and the other temperature scale height with the mean free path, but a uniform language should be chosen. As to my initial question, the heat flux, that is strictly speaking simply the third momentum of the random speed, is considered large as compared to mechanical flux; however, intuitively this seems at odd with a negligible transfer of momentum between particles, i.e., with the fact that the system is collisionless (i.e., large mean free path). For a Nature paper that should reach out various communities, a distinction between the heat flux and momentum exchange could help and a clarification why one is large and the other negligible, could help.

About my previous point 2), I am not sure whether or not the answer means that authors expect that 3D-RTI would be affected more, or to the same extent of 2D, by the heat flux in SN.

After those clarifications the paper should be publishable.

Reviewer #2 (Remarks to the Author):

I have read over the revised manuscript and believe that the authors have adequately responded to the reviewers' comments. Publication of the paper is warranted.

A few typos:

p. 2 line 6 supernovae -> supernova

p. 2 line 13 in understanding -> in the understanding

p. 9 middle star -> SNR

Reviewer #3 (Remarks to the Author):

The article is now much clearer. Most of the doubts have been resolved, but I still think the explanation of fig 4 is too speculative. I think the article is suitable for publication.

Reviewer #1 (Remarks to the Author):

I would like to thank the authors to carefully revise the paper and answer my questions. However, I would like a clarification about some points, not yet clear in my opinion.

About my previous point 1), I think consistency in definition and terminology would help clarity. Authors report in table 1 that the performed NIF experiment is as much collisional as the SN 1993J. If laboratory experiments are usually considered collisional because of the extremely large electron density, on the other hand astrophysical shocks, and so shocked layers in SN, are typically collisionless: Coulomb collision mean free path is much smaller than the typical scale of the system. Table 1 gives a very small λ_c (that is not defined but is presumably such mean free path).

In their reply, the authors define the shocked layer collisionless because the mean free path is much larger than the temperature scale height, whose definition I could not find. So a consistency would be appreciated.

The authors are merging the language of two separate communities, one comparing L and the other temperature scale height with the mean free path, but a uniform language should be chosen.

As to my initial question, the heat flux, that is strictly speaking simply the third momentum of the random speed, is considered large as compared to mechanical flux; however, intuitively this seems at odd with a negligible transfer of momentum between particles, i.e., with the fact that the system is collisionless (i.e., large mean free path). For a Nature paper that should reach out various communities, a distinction between the heat flux and momentum exchange could help and a clarification why one is large and the other negligible, could help.

Indeed we were insufficiently precise in our discussion of collisionality and in the various behavior of the ions and the electrons. Improving this aspect led us, in the main text, to specify λ_c as the mean free path for ion-ion collisions:

Both systems have characteristic length $L \gg \lambda_c$, the mean free path for ion-ion collisions, in their denser shocked layers.

and to replace “are highly collisional” with “have collisional ions” near the bottom of page 7.

These changes in the main text reflect the requirement that the ions be collisional in order to have fluid-like behavior. In the supplemental material, we now state

“Whether the plasma behaves as a fluid is determined by the ion-ion collisionality, and so we have evaluated the collisional mean-free path in the main text as the mean free path for ion-ion collisions, $\lambda_c = v_i / \nu_{ii}$, in which v_i is the ion thermal velocity and ν_{ii} is the ion-ion collision frequency. The electrons are nearly collisionless in the shocked CSM, with implications we just discussed, but are collisional in the cooler, denser shocked ejecta. And the ions, with their shorter mean-free path, are collisional in both regions.”

With regard to the question of the referee about the relation of the heat flux to momentum exchange, we note that the heat in the systems of interest is carried by the electrons, and we agree that the heat flux is the (vector) third moment in velocity of the electron distribution function. The ions, however, carry a much smaller heat flux. In contrast, the momentum is almost all carried by the ions and it is ion-ion collisions that drive the hydrodynamic behavior. The main effect of momentum exchange between electrons and ions is to randomize the motion of the electrons, and to gradually heat or cool them depending on the difference between electron and ion temperature. We hope that the referee agrees with us

that our improved discussion the collisionality of the various species and the origins of fluid behavior, described above, eliminates the need for a specific discussion of momentum exchange.

About my previous point 2), I am not sure whether or not the answer means that authors expect that 3D-RTI would be affected more, or to the same extent of 2D, by the heat flux in SN.

Apologies, we misunderstood your previous comment. We do believe that 3D RTI will be more affected by the heat flux, however, we do not have clear evidence of this and therefore did not put it in the publication.

After those clarifications the paper should be publishable.

Reviewer #2 (Remarks to the Author):

I have read over the revised manuscript and believe that the authors have adequately responded to the reviewers' comments. Publication of the paper is warranted.

A few typos:

p. 2 line 6 supernovae -> supernova

p. 2 line 13 in understanding -> in the understanding

The above has been fixed.

p. 9 middle star -> SNR

I cannot find an occurrence of "middle star" in the paper

Reviewer #3 (Remarks to the Author):

The article is now much clearer. Most of the doubts have been resolved, but I still think the explanation of fig 4 is too speculative. I think the article is suitable for publication.

In addition to the changes discussed above we have also replaced the right side of Figure 3 with simulation results of mixed width vs. tau plot. Previously, we have the spike or bubble height from the mean interface. However, after further discussion amongst the authors we believe that the mean interface position can be misleading since the position of the interface itself is affected by the ablation. Future publications by the authors are being prepared to discuss this further as this is beyond the scope of this publication.

REVIEWERS' COMMENTS:

Reviewer #1 (Remarks to the Author):

I would like to thank the authors to further clarifying the requested points.

One last point I have is that in Suppl Mat authors should also report, in the paragraph before last at page 8, which expression they use for ν_{ii} , ion-ion collision frequency, and which temperature (whether or not ions and electrons are in equilibrium), in the shocked and unshocked regions. Also, the ratio λ_c/L in table 2 has changed from previous version. It was the same for NIF and SNR, but different by a factor 10^{-4} in the current version. Is this because previously a different λ_c was used?

Reviewer #1 (Remarks to the Author):

I would like to thank the authors to further clarifying the requested points. One last point I have is that in Suppl Mat authors should also report, in the paragraph before last at page 8, which expression they use for ν_{ii} , ion-ion collision frequency, and which temperature (whether or not ions and electrons are in equilibrium), in the shocked and unshocked regions.

We have added the definition of ν_{ii} and specified the temperature used.

Whether the plasma behaves as a fluid is determined by the ion-ion collisionality, and so we have evaluated the collisional mean-free path in the main text as the mean free path for ion-ion collisions is $\lambda_c = v_i/\nu_{ii}$, in which v_i is the ion thermal velocity and ν_{ii} is the ion-ion collision frequency where

$$\nu_{ii} = 4.80 \times 10^{-8} Z^4 n_i \ln \Lambda T^{-3/2} \text{sec}^{-1} \text{ and}$$

we assume that $T_i = 10T_e$.

Also, the ratio λ_c/L in table 2 has changed from previous version. It was the same for NIF and SNR, but different by a factor 10^{-4} in the current version. Is this because previously a different λ_c was used?

In the previous version λ_c was calculated incorrectly. In the current version it is calculated as expressed in the text.